# Overview and Challenges of Large-Scale Cultivation of Photosynthetic Microalgae and Cyanobacteria

**DOI:** 10.3390/md21080445

**Published:** 2023-08-10

**Authors:** Lucie Novoveská, Søren Laurentius Nielsen, Orhan Tufan Eroldoğan, Berat Zeki Haznedaroglu, Baruch Rinkevich, Stefano Fazi, Johan Robbens, Marlen Vasquez, Hjörleifur Einarsson

**Affiliations:** 1ScotBio Ltd., Livingston EH54 5FD, UK; 2Ocean Institute, 1201 Copenhagen, Denmark; 3Department of Aquaculture, Faculty of Fisheries, Cukurova University, 01330 Adana, Türkiye; 4Institute of Environmental Sciences, Bogazici University, 34342 Istanbul, Türkiye; 5National Institute of Oceanography, Haifa 31080, Israel; 6Water Research Institute, National Research Council of Italy (IRSA-CNR), 00015 Roma, Italy; 7Flanders Research Institute for Agriculture, Fisheries and Food, 9820 Merelbeke, Belgium; 8Department of Chemical Engineering, Cyprus University of Technology, Limassol 3036, Cyprus; 9Faculty of Natural Resource Sciences, University of Akureyri, 600 Akureyri, Iceland

**Keywords:** large-scale cultivation, microalgae cultivation, cyanobacteria cultivation, scale-up

## Abstract

Microalgae and cyanobacteria are diverse groups of organisms with great potential to benefit societies across the world. These organisms are currently used in food, feed, pharmaceutical and cosmetic industries. In addition, a variety of novel compounds are being isolated. Commercial production of photosynthetic microalgae and cyanobacteria requires cultivation on a large scale with high throughput. However, scaling up production from lab-based systems to large-scale systems is a complex and potentially costly endeavor. In this review, we summarise all aspects of large-scale cultivation, including aims of cultivation, species selection, types of cultivation (ponds, photobioreactors, and biofilms), water and nutrient sources, temperature, light and mixing, monitoring, contamination, harvesting strategies, and potential environmental risks. Importantly, we also present practical recommendations and discuss challenges of profitable large-scale systems associated with economical design, effective operation and maintenance, automation, and shortage of experienced phycologists.

## 1. Why Cultivate Microalgae and Cyanobacteria?

### 1.1. Age of Algae and Novel Compounds

Microalgae and cyanobacteria are very diverse groups of organisms with under-explored and under-valued potential. There are thousands of species that thrive in marine, freshwater, and even terrestrial environments. A conservative estimate is that there are 44,000 scientifically described microalgal and cyanobacterial species [1]. This vast number of species naturally encompasses diversity in biochemical compounds as well. Environmental conditions also alter the biochemical composition and trigger the production of certain compounds, which means that the total biochemical potential for commercial exploitation is very high [2]. In this review, we evaluate the cultivation of both microalgae and cyanobacteria. These two groups are categorically different (e.g., microalgae are eukaryotic while cyanobacteria are prokaryotic). However, the cultivation principles and challenges overlap. Moreover, the valorization potential and the valorization routes are very similar between both groups. In addition, approaches to performing metabolic engineering are similar [3] (e.g., production of added value compounds), as well as application and features of accumulated biomass.

Microalgae and cyanobacteria are both groups of organisms with the capacity to perform photosynthesis. Therefore, from a technical perspective, both groups can be grown in photobioreactors and ponds, however often with a variation in specific cultivation conditions. Cyanobacteria, being prokaryotes, have relatively simpler nutrient requirements, and some species can fix atmospheric nitrogen, reducing the need for external nitrogen sources. However, condition requirements, growth parameters, downstream processes, etc., also differ significantly between species rather than between cyanobacteria and microalgae as groups.

Microalgae and cyanobacteria are cultivated for several products used in food, feed, nutraceutical/pharmaceutical and cosmetic industries. These products range from bulk components such as protein, carbohydrates, and lipids to more specific compounds that are present in much smaller concentrations but may have a significant value. These more specific compounds include pigments and colorants, long-chain polyunsaturated fatty acids (n-3 LC-PUFAs) commonly known as omega-3 fatty acids, phytohormones, enzymes, anti-oxidants, anti-fungal, anti-microbial, anti-virus, anti-inflammatory, anti-cancer and anti-obesity compounds, peptides, vitamins, and iodine [4]. In this sense, with the inclusion of algal species in aquafeeds, more than twenty species have been previously assessed, with well-documented benefits across a variety of fish species [5]. Indeed, trace elements (calcium, magnesium, phosphorous, iron, iodine, and zinc) are abundant in microalgae, and these health-promoting compounds make microalgae suitable for use as natural supplements in aquafeed. In particular, extremely high iodine concentrations present in the microalgae showed that these algae have a high ability to accumulate iodine [6]. Hence, presently these microalgal products are globally considered future sources of feed and novel food ingredients with the most potential. It is also obvious that microalgae as single-cell sources will soon constitute a realistic potential alternative or addition to animal feed globally. 

Currently, commercial cultivation is dominated by the production of pigments, n-3 LC-PUFAs, and proteins. Large-scale cultivation of microalgae and cyanobacteria is a small industry worldwide, but the interest and number of biotech companies are rising. The European Algae Biomass Association (EABA) estimated the economic value of the microalgae sector to be around €850 million/year [7]. 

### 1.2. Microalgae for Production of Aquaculture as Live Feed and Feed Ingredient

Aquaculture, the fastest-growing animal production sector, is the driving force in the development of new animal protein sources as wild fisheries exceed their sustainable limits. In this regard, microalgae are mainly used for their nutritional value [8], and the current potential utility of these ingredients is considerable and growing [9]. 

It is important to distinguish between the use of microalgae as a live feed or as a component of formulated feed, e.g., salmonids and crustaceans. In a feed context, microalgae are used both directly as feed for the cultured organisms or, in some cases, indirectly as feed for feed organisms, typically invertebrates. By far, the most important use of microalgae for direct feed in terms of volume and value is for filtering mollusks—oysters, clams, mussels, and pectinids, followed by shrimp farming [10]. The direct use of microalgal feed to fish is limited to herbivorous plankton feeders, especially cyprinids and cichlids [10]. When microalgae are used as indirect feed organisms in aquaculture, they are primarily fed to rotifers and Artemia, which in turn are fed to fish. Many fish species, particularly marine finfish (excluding salmonids), rely on live feed in their larval stages [11]. However, neither rotifers nor Artemia are perfect as live feed in fish production, mainly because they do not have the right composition of n-3 LC-PUFAs, necessitating the use of enrichment of the feed organisms [11]. Another possibility is to use copepods with a composition of LC-PUFAs and other essential nutrients more suitable for fish feed [12]. Copepods are also fed microalgae [13]. 

Only a few species have gained widespread use in aquaculture, with the most popular ones being *Tisochrysis* spp., *Diacronema lutheri*, *Tetraselmis suecica*, *Nannochloropsis* spp., and *Chaetoceros* spp. [14]. *Dunaliella* and *Haematococcus* are also among the successful products based on their long-chain isoprenoid compounds, e.g., carotenoids [15,16].

Microalgae species for use in aquaculture must have rapid growth rates, balanced nutrient composition, and be easy to culture in large-scale facilities [14]. Within the aquaculture sector, the main focus is on fatty acids, especially n-3 LC-PUFA, amino acids, carotenoids, and vitamins [17]. The LC-PUFA profile of microalgae is under both genotypical and phenotypical control and so varies both among taxonomic groups and can be controlled, at least partly, by manipulating the growth conditions of the microalgae as well [18]. The vitamin content of different microalgae also varies significantly across taxonomic groups of algae, while the amino acid content and composition are much more constant [14]. 

In recent decades, some current investigations have reported the feasibility of sustainable microalgae production, both whole-cell and processed form used as feed ingredients formulating industrially compounded aquafeeds for salmonids, shrimp, tilapia, and many other marine species [9,19,20]. It is obviously seen that microalgae can be used in the feed industry as feed sources; however, the feed industry demands raw materials available in large volumes. With this regard, it is commercially important to produce microalgae in large-scale and lower-cost production methods.

### 1.3. Bioremediation

Research so far demonstrates that single-cell organisms (e.g., microalgae and cyanobacteria) can develop rapidly on a range of organic and inorganic inputs providing an economically rewarding and environmentally friendly strategy for reducing waste volume. Microalgae and cyanobacteria are effective at absorbing nutrients (nitrogen and phosphorus), carbon, and other contaminants (e.g., heavy metals and pharmaceutical compounds) present in their surroundings. These properties make them a suitable candidate for a phycoremediation. As the cells multiply, they consume nutrients and contaminants, which is effectively treating wastewater [21]. In addition, daytime photosynthesis produces dissolved oxygen allowing for effective biological oxygen demand (BOD) removal [22]. Bioremediation is typically performed with polycultures or entire consortia of a variety of microbes. Large-scale wastewater treatment (WWT) is an increasing commercial venue [23]. Integrating wastewater treatment with algae and cyanobacteria production provides cumulative benefits of water recycling, eliminating the need for external fertilizer while providing efficient wastewater treatment.

## 2. Species Selection

Phycologists must first decide on the target product or application and select the most suitable species or combination of species accordingly. As discussed earlier, the biochemical composition is known only in a relatively small number of species, and more research is needed. Species and strains may be isolated from their natural habitats or obtained from culture collections. Phycologists may perform bioprospecting first to screen several species for the best yield and best-growing conditions of the targeted compound [24]. The most frequently used commercial genera are *Haematococcus*, *Dunaliella*, *Chlorella,* and *Arthrospira* (commonly called Spirulina). Other commonly used species are *Nannochloropsis* spp., *Isochrysis* spp., *Thalassiossira* spp., *Tetraselmis* spp., *Chaetoceros* spp., etc. The number of microalgae-related applications to the EU Novel foods list (see the website) is slowly increasing. There are three additional microalgal species or their extracts (*Phaedactylum tricornutum*, *Tetraselmis chuii*, and *Galdieria sulphuraria*) on the application list. Application for Novel food regulation in Europe via European Food Safety Authority (EFSA) can be an expensive and long journey. However, the Qualified Presumption of Safety (QPS) status of some microalgae, as assigned by the EFSA, is a fast track to approval. In Europe, approval as Novel Food is needed for food that has not been consumed by humans in the EU before 15 May 1997. The Novel Food regulation came into force as Regulation (EU) 2015/2283. ‘Novel Food’ can be newly developed, innovative food as well as food produced via new technologies and production processes or food that is mainly (traditionally) eaten outside of the EU. In the USA, the GRAS regulation (Generally Recognized as Safe) is in place and allows several food ingredients on the market (e.g., *Haematococcus pluvialis* extract containing astaxanthin and the dried biomass of *Arthrospira platensis* [25]. Spirulina (*Arthrospira*) and *Nostoc* were already on the market before 15 May 1997. Spirulina has been used as a food source for hundreds of years in Mexico (by Aztecs) and Africa. *Nostoc* is a filamentous cyanobacterium traditionally consumed in China, Mongolia, Tartaria, and South America. 

### 2.1. Genetic Modification

#### 2.1.1. Genetic Adaption of Cells for Mass Production

Desirable properties of the selected species can also be enhanced via targeted genetic engineering or indirect mutagenesis approaches. Large-scale cultivation of genetically modified organisms (GMOs) remains a controversial and difficult topic in most countries, including the EU, and GMO mass commercial cultivations are still rare. Microalgae are eukaryotic organisms with complex genetic makeup, and the number of model species is still lacking. However, the number of genome-sequenced species is on the rise, and genetic engineering tools such as micro-RNA (miRNA), silence RNA (siRNA), and CRISPR-Cas technologies have already been applied to algae [3]. 

Meanwhile, genetic modification (GM) is much more robust for cyanobacteria, an advantage of being prokaryotic cells with increased transformant efficiency. Many algal biorefinery high-value compounds have been globally gaining momentum through genetic engineering by improving the mass cultivation of algae and cyanobacteria [26,27,28,29]. However, comprehensive approaches need to be addressed to achieve a thriving algal biorefinery with high efficiency.

GM efforts to improve photosynthetic efficiency for mass cultivation are also popular for the dual benefit of increased biomass productivity and higher titers for desired products unless a stress mechanism interferes with overall accumulation and biomass productivity. Adjusting the chloroplast antenna size is one of the most common GM methods to sustain high biomass productivity at a large scale, especially for products requiring higher light intensities [30,31]. GM of light-harvesting complex to increase capture efficiency even at low light conditions also contributes to higher product yields. Beneficial for large-scale applications, one particular example has shown that extending the absorption of photosynthetically active radiation (PAR) region from 700 nm to 750 nm increased the available photon amount by 19% [32].

#### 2.1.2. Engineering to Increase Chemical Composition and Yield

For most GM applications, enzyme activity controlling the synthesis of bioproducts from microalgae and cyanobacteria is targeted to overcome rate-limiting steps, over-express synthesis, and decrease catabolic reactions. Controlling flux in competing pathways is also another strategy to achieve desired bioproduct yield. In our review of the literature, we found various bio-engineering applications to produce and increase the target compounds present in microalgae and cyanobacteria. For instance, the growth condition and strain selection can greatly affect cyanobacteria polyhydroxyalkanoates (PHAs) composition, polymers being used to generate biodegradable plastic [33]. In another case, RuBisCO, often considered an inefficient enzyme due to relatively low turnover rates in CO_2_ fixation at the core of the Calvin Cycle, can be can successfully engineered to boost photosynthetic efficiency in both microalgae and cyanobacteria [34]. Interestingly, targeted mutagenesis and/or overexpression of some of the genes reduced fatty acid toxicity and subsequently led to an increase in fatty acid production [27,29]. In biobutanol production, Shanmugam et al. [35] provide information about genome engineering techniques for the direct conversion of CO_2_ into biobutanol using solar energy harnessed by microalgae. 

So far, significant advances in the development of genome engineering techniques (such as RNAi, ZFNs, TALENs, and CRISPR-Cas9) provide functional insight into genes and relevant proteins involved in central carbon metabolism [26,28,35]. Recently, Artificial Intelligence (AI)-assisted screening of microalgae species (selected based on high biomass production) and strains (selected based on commercial application in feed, nutrition, or cosmetics) provide pertinent information for the genome sequence, which can assist in optimization, detection, and alteration of gene expression. AI-based genomic bioinformatics was reviewed very recently by Teng et al. [26].

### 2.2. Genetic Engineering to Increase Growth under Limiting Conditions

Recent advancements in next-generation sequencing tools have also contributed to the algae and cyanobacteria biotechnology field especially high-throughput data obtained under several limiting cultivation conditions. Particularly RNA-Seq-based transcriptomics help unravel the expression of key synthesis genes under certain stress conditions. A study by Li et al. on *Nannochloropsis oceanica* strain IMET1 identified how nitrogen-limited conditions affect the flux of protein and carbohydrate pathways diverted onto lipid synthesis pathways and improved triacylglycerol (TAG) accumulation [29]. In another study by Chang and colleagues, the downregulation of triosephosphate isomerase led to improved fatty acid and starch synthesis in *Neodesmus* sp. UTEX 2219-4 [36]. Despite the improved metrics on overall lipid accumulation (especially for biofuel applications), nutrient stress conditions cause lower biomass yields and remain a challenge to be addressed.

## 3. Three Most Common Ways of Large-Scale Cultivation

Microalgae and cyanobacteria are among the most promising resources to provide multiple energy and environmental benefits, yet, the low productivity and high cost of algal cultivation impede advancement in their intensive applications. The state-of-the-art photosynthetic algae cultivation approaches have primarily focused on algal cultures in open ponds. Although these systems are easily built, they are susceptible to light limitations and stresses that hamper algal growth beyond a cell concentration of 0.5 g/L in open ponds and 2–6 g/L in photobioreactors [37]. Moreover, biomass harvesting and concentration are extremely costly due to these low algal cell densities [38]. There are systems, such as thin-layer cascade photobioreactors, that can reach a biomass density of more than 30 g/L [39]. However, these high-density systems face other challenges, such as light limitations. A similar system, such as Seeded Algal Turf Scrubber (sATS), can be used to grow different types of algae [40].

The mass production of microalgae and cyanobacteria can be divided into three main areas of operations: ponds, photobioreactors, biofilm systems, or a combination of all. The challenges and advantages of each system are summarized in Table 1, and examples of large-scale cultivation systems are shown in Figure 1. While ponds are relatively simple and inexpensive compared to photobioreactors, they require large land area, they are ineffective in CO_2_ efficiency, they are exposed to possibly unfavorable weather (rain, evaporation, etc.), and they can get contaminated easily, which is a drawback for certain applications such as food and feed. On the other hand, photobioreactors (PBRs) are more robust and significantly more expensive in CAPEX and OPEX. PBRs are built with a high ratio of surface area to culture volume to maximize illumination. However, the same principle allows heat loss in the absence of light. Understanding time-dependent heat balance within the PBR is critical for accurate productivity predictions [41]. Biofilms are a unique way to cultivate microalgae and cyanobacteria, which are relatively inexpensive, require very little or no dewatering, and are often dominated by polyculture and complex consortia, including bacteria that may impact the biochemical composition of the final harvested biomass. 

There are many factors to consider when selecting the most suitable cultivation systems. These include species preference, application (cultivating for products versus bioremediation application), productivity target, land and water availability, climate, accepted contamination risk, CAPEX, and OPEX [42].

### 3.1. Ponds

In general, ponds are considered the cheapest method for microalgal production, both in terms of construction and operation. The simplest and most primitive method of producing microalgae in connection with aquaculture is the so-called green ponds or green-water ponds. In these ponds, dense microalgae blooms develop because of fertilization with chemical or manure fertilizers or simply because of nutrient release from feed utilization. The algal blooms contribute through a zooplankton food chain to the nutrition of larvae of fish or shrimp. In these systems, there is no control of the microalgae population and hence no ability to culture a specific microalgae species with a specific nutritional value [43,44]. In addition, green-water systems crashes, culture upsets, contaminations, and seasonal variability are common. 

A more advanced option for microalgal production is to construct ponds specifically for this purpose. These are most often shaped as elongated ovals, lending them the name raceway ponds. Other shapes are also used, although less frequently. Even though constructed ponds are cheaper to construct than, e.g., PBRs [45], they still have a much larger physical footprint than these and take up larger land areas [46]. Several factors contribute to making production in raceways less efficient than in other types of industrialized microalgal production. 

First of all, self-shading is a bigger problem in ponds than in PBRs because ponds have longer light paths than PBRs [46]. This is necessary for technical reasons: Ponds need a certain depth to avoid too rapid evaporation and for paddle wheels to operate efficiently. They are usually 20–40 cm deep [46]. In contrast, PBRs can have very short light paths depending on tube diameter or panel thickness of 5–6 cm is standard, but even shorter light paths have been used [47]. The longer light paths and the large self-shading mean that ponds must operate with lower algal cell densities than PBRs, giving them lower volumetric production rates. Raceway pond volumetric production rates are usually in the range of 0.010–0.12 g L^−1^ d^−1^, while PBRs have volumetric production rates in the range of 1.5–1.6 g L^−1^ d^−1^ [46,48]. In a recent review, the volumetric productivity of various microalgae was reported from 0.01 to 0.42 g L^−1^ d^−1^ [49]. Another review reports higher productivity or 0.1 to 3.8 g L^−1^ d^−1^ for microalgae and 2.7 g L^−1^ d^−1^ for *Arthrospira* (Spirulina) [50]. Similar values are reported by Chen et al. [51], demonstrating productivities from 0.01 to 7.4 g L^−1^ d^−1^.

Comparison of different photobioreactor configurations shows that for the same species (i.e., *Nannochloropsis*), biomass yield in raceway pond was lower 0.03–0.2 g L^−1^ d^−1^ (6.2 to 24.5 g m^−2^ d^−1^) compared to 0.12 to 0.85 g L^−1^ d^−1^ in horizontal tubes, 0.31 to 1.45 g L^−1^ d^−1^ in vertical tubes and 0.15–1.2 g L^−1^ d^−1^ in flat panel PBR [52]. In addition, mixing is generally less efficient in ponds than in PBRs, so it is usually necessary to install bafflers, etc., in the pond to improve water mixing and prevent the formation of an optical dark zone [53]. Finally, open ponds are prone to contamination and/or predation from other organisms, contributing to their low efficiency [54]. The latter factor may be prevented if the production is focused on microalgae tolerating extreme environments, such as high salinity, high or low pH, etc., in which case contamination and predation are limited due to the extreme environment in the ponds [55].

### 3.2. Photobioreactors for Photosynthetic Algae and Cyanobacteria

#### 3.2.1. Tubular Photobioreactors

It is a known fact that the commercial value of bioproducts obtained from microalgae and cyanobacteria often require axenic cultivation systems with higher biomass productivity. As a result, enclosed PBRs are preferred for large-scale cultivation with improved CO_2_ capture and minimized contamination issues. Meanwhile, enclosed PBRs are more expensive compared to raceway ponds; therefore, economically feasible designs are popular such as transparent, cheaper plastic tubular bags with short replacement periods or more durable glass tubular PBRs. Tubular PBRs have the advantage of placement in various orientations, vertical, horizontal, tilted, etc., and are aligned for optimal capture of natural and/or artificial light. For large-scale operations, tubular diameters of 10 to 50 cm with recirculating culture are preferred to overcome the limitation of light diffusion as cultures mature, and the length is adjusted based on system aeration, degasser, and flow conditions to ensure well-mixed culture. 

Troschl and colleagues have recently cultivated *Synechocytis* sp. strain CCALA192 in a 200-L tubular photobioreactor for poly-β-hydroxybutyrate (PHB) production over 75 days of production cycle and achieved 1.0 g/L biomass concentration with maximal PHB concentration of 12.5% dry cell weight (DCW) [56]. *Tetraselmis* sp. CTP4 was successfully scaled up from an agar plate to 35 and 100 m^3^ industrial-scale tubular PBRs by Pereira et al. in semi-continuous cultivation for 60 days during autumn-winter conditions where 0.08 g L^−1^d^−1^ volumetric and 20.3 g m^−2^d^−1^ areal biomass productivities were obtained attained in the 100 m^3^ compared to those of the 35 m^3^ PBR (0.05 g L^−1^d^−1^ volumetric and 13.5 g m^−2^d^−1^ aerial productivities) [57].

For large-scale applications, there are also examples of hybrid tubular systems. García-Galán and colleagues have successfully demonstrated an 8.5 m^3^ capacity semi-closed (hybrid) tubular horizontal photobioreactor (HTH-PBR) for the treatment of agricultural runoff [58].

#### 3.2.2. Column Photobioreactors

Column PBRs are vertical tubes of varying diameters and heights. Column PBRs are relatively simple to make, easy to operate, and are regarded as cost-effective. However, the volume per column is limited. They are usually made of rigid transparent plastic material, although glass is sometimes applied. Vertical plastic bags can be considered as one type of column PBR. The diameter is restrained by the length of the light path and can vary from a few cm up to 30 cm or more, where 20 cm has been recommended as the maximum diameter. The height of the columns is mainly restrained by the pressure buildup in the column, and the maximum height is 3–4 m. Column PBR has a favorable volume-to-floor space ratio. Column PBR can be difficult to clean but is considered easy to sterilize [59,60,61]. 

Mixing is usually obtained by pumping in the air from the bottom, through a gas disperser and nozzles, in a bubble column or airlift configuration. The air can be mixed with CO_2_ or other gases. The airlift configuration can be formed in different ways, e.g., as a split column or as an internal loop [62,63]. 

Column PBR can be illuminated by sunlight or, more preferably, illuminated by artificial (electrical) lights, e.g., LED. If placed outside, the columns can be tilted to achieve effective light-angle [62].

In order to scale up, several columns can be added either as separate units or connected. Separate units can be controlled individually, and contamination is more likely to be limited to one or a few columns. Using the dimensions given above, a typical column can contain up to 400 L; however, a volume of 300 m^3^ has been used [64].

#### 3.2.3. Indoor Production in Open Containers

While the production of microalgae for aquaculture in developing countries often takes place in green water ponds as described above [44], the most common method for large-scale production of microalgae for aquaculture in industrialized countries is the use of transparent, disposable, plastic bags [54]. The main advantage of this method is that it is cheap. The bags themselves are cheap, and this type of operation does not require very complex installations. They just require a support system to vertically hang the bags from, light, a system for aeration/mixing the bags with air with or without CO_2_ addition, and a system that allows filling and emptying the bags. The main disadvantage is that production in bags is, by definition, batch production—it does not allow for continuous microalgae production. Hence, it is necessary to maintain production in a number of bags containing microalgae in different stages of the production cycle. Since each bag is a separate batch production unit, economies of scale require that the units be large, with a large diameter and a large optical path. Therefore, the bags’ optimal cell density is small, resulting in a low cell density (cells L^−1^) and low volumetric productivity [54].

### 3.3. Biofilms

Biofilms are formed by the attachment of microorganisms on submerged surfaces in the aquatic environment. Biofilms in nature are complex microbial communities, continuously exposed to a diverse inoculum that includes bacteria, archaea, algae, fungi, protozoa, and even metazoa. Biofilm-based algal cultivation has received increased attention because of three major advantages: i. resistance to growth stresses, ii. high cell density, iii. low harvesting and concentration costs [38]. In addition, algal biofilm systems also have a variety of unique features, such as minimization of light limitation and enhancement of CO_2_ mass transfer. Separation of the algal cells and liquid medium also increases the retention time of the cells without being washed out [65]. Algal biofilm reactors were first reported in the 1980s used for removing nitrogen and phosphorus from municipal wastewater. To date, progress has been made in using microalgae biofilm for wastewater treatment and biomass production [38].

The algal biofilm system has its own unique factors that affect overall performance, including algal species, the properties of attachment material, the algal biofilm thickness, and the shear stress applied to the biofilm. Biofilm’s formation and growth depend on cell-surface properties, particularly pertaining to surface hydrophobicity and surface roughness/pattern. The selected materials for algal cell attachment need to be durable, inexpensive, easy to be obtained, and non-toxic to algal cells (cotton rope, membranes, stainless steel woven meshes, etc.) [66]. Some biofilm systems use attachment materials with many crevasses that create additional attachment areas, but this additional area does not have access to light. This system is commonly used in wastewater treatment [67], with a consortium of algae and bacteria involved [68]. Other biofilm systems use flat attachment materials that are designed to allow maximal light exposure to the biofilm. The biomass in these systems can be easily harvested [65].

Algae biofilm systems can be designed with multiple layers (horizontal, vertical, and rotating design), increasing the productivity per land area and the efficiency of land use [38]. A comprehensive summary of the different algal biofilm systems is provided by Gross et al. 2015 [65]. The authors categorized the different types of algal biofilm systems based on the relative movement of the algal biofilm to the liquid medium, e.g., stationary biofilm or rotating biofilm.

## 4. Water and Nutrient Sources

### 4.1. Seawater as Growth Medium

Seawater media can be obtained either from mixing the basic salts of seawater, from mixing commercial salts in either tap water or deionized water, or from natural seawater. There are a number of well-established recipes for artificial seawater available in the scientific literature [69], and mixing artificial seawater from the basic ingredients obviously gives full control of the composition of the medium. It is, however, a rather expensive and time-consuming approach for the production of large volumes of media. For this purpose, the use of commercial salt may be preferred. There are a number of various brands available [70] that can be mixed with either tap water or deionized water with little or no difference in their suitability for microalgal production [70]. However, it must be emphasized that the intended use of most of these commercial salts is for the use in keeping ornamental fish, not producing microalgae. Some commercial salts may therefore be deficient in certain nutrients and may require the addition of extra micro- or macro-nutrients [71]. The use of natural seawater may be preferred when available since using natural seawater in most cases will make the addition of micronutrients and vitamins unnecessary. The addition of macronutrients such as nitrogen, phosphorus, and potassium may, however, still be required.

### 4.2. Use of Agricultural Fertilizers as Nutrient Source in Microalgal Production

In the scientific literature, there are many recipes available for microalgae growth media [72]. Some are generic and may be generally used for the production of many different groups of microalgae; some are more specialized for specific groups of microalgae, taking into account their need for particular nutrients or growth factors. They have in common that they are expensive and time-consuming to use in large-scale microalgal production because they rely on laboratory-grade chemicals and laboratory procedures. Because of this, there has been an interest in testing the use of agricultural fertilizers for large-scale microalgal production, as these are relatively cheap compared to laboratory-grade chemicals and are available in large quantities. It is generally found that the use of agricultural fertilizers, such as those used in farming in general, yields production rates of biomass equal to or even better than traditional media [73,74,75] while at the same time providing equal or better production yields of the required end product [76]. This means that the use of agricultural fertilizers in microalgal production may reduce the costs of the growth medium by as much as 95% [77,78]. However, when using agricultural fertilizers, it should be kept in mind that they are meant for use in a soil matrix and, therefore, may be deficient in some micronutrients or trace elements. On the other hand, they may contain trace elements that potentially can accumulate to levels toxic for the microalgae.

### 4.3. Use of Wastewater as a Nutrient Source

Municipal wastewater (WW) is any water that has been contaminated by human use. WW is a complex matrix containing significant concentrations of solids, dissolved and particulate matter, microorganisms, nutrients, heavy metals, and micro-pollutants. Wastewater treatment (WWT), mainly implemented in the so-called activated sludge processes, is used to remove these contaminants from wastewater and convert it into an effluent that can be returned to the environment. WWT is an opportunity for microalgae production as providing satisfactory levels of organic/inorganic nutrients [79]. In this context, microalgae have a vital role in WWT in facultative systems, mainly high-rate algal ponds, to minimize the energy input and to utilize nutrient-rich effluent streams in WWT [80]. Microalgae production can thus be combined with WWT and industrial sources of organic carbon, inorganic phosphorus (P), or nitrogen (N, including ammonia) as nutrient removal [80,81,82]. However, algal production and recovery of wastewater nutrients are dependent on critical variables such as pH, temperature of the growth medium, macronutrients, micronutrients, and nutrient ratio (e.g., C:N:P). It is shown that *Chlorella* and *Scenedesmus* are effectively used in municipal wastewater in oxidation ponds and in high-rate algal ponds and have both been shown to grow in a broad range of wastewaters [80,82,83]. Moreover, there is interest in using microalgae for the remediation of industrial-derived wastewater containing metal pollutants (cadmium, chromium, zinc, etc.) rather than N and P [84,85]. However, there is limited use for the accumulated biomass. If microalgae and cyanobacteria are grown on wastewater feeds, the final biomass may contain a variety of contaminants and pathogens. This biomass is unsuitable for human consumption and is predominantly used for the production of biofuels [86].

## 5. Temperature Regulations

### 5.1. Outdoor Temperature and Climate

Even though there are psychrophilic (cold-loving) and psychrotolerant algal species, the majority are mesophilic with optimal growth temperatures at 10–25 °C. Thermophilic algae are relatively uncommon, and the majority of algal cultures crash above 40–45 °C. For large-scale outdoor cultivation, solar intensity levels at the production sites are vital as high intensities may create a risk for overheating during summer months, while lower intensities may not be sufficient to provide enough heat during cold seasons [63]. Another important issue for temperature regulation for large-scale operations is the diurnal fluctuations. Dessert/dessert-like climates usually experience high temperatures during the day, while temperatures drop sharply during the night, reaching more than 20 °C differences for outdoor cultivation systems.

With extended surface areas, open ponds have a certain tolerance to temperature variations, even though ponds readily suffer volume losses via evaporation in hot climates. Outdoor cultivation systems operated under natural sunlight may need to be supported with cooling and/or heating systems depending on the climate conditions of the cultivation site. These systems lead to higher CAPEX and OPEX, though some low-cost strategies, such as water spraying and shading systems, can be implemented. Circulating chilled or heated water within or around the cultivations systems are also widely implemented solutions for controlled temperature. There are also examples of submerging PBRs in larger ponds [87].

Another novel application that can be applied for temperature regulation for large-scale operations is designing PBRs with phase change materials (PCMs). By phase transitioning, PCMs basically absorb or release energy and enable control of heating/cooling. Among more than 150 different PCM, low-cost, non-corrosive, and non-toxic materials can be implemented to minimize the energy requirements used for active cooling and/or heating of large-scale PBR systems. Uyar and Kapucu have successfully demonstrated temperature control for 80 L capacity PBRs operating under sunlight with PCMs such as capric acid [88].

### 5.2. Indoor Temperature and Conditions

Indoor production of microalgae is more expensive than outdoor production as it requires investment in and maintenance of buildings. Furthermore, it is associated with expenses for artificial lighting and temperature control—heating and/or cooling. However, if the value of the product is sufficiently high, indoor production may be justified. Products of sufficiently high-value range from microalgae to be used in live feed aquaculture operations [89] and upwards in the market value pyramid. The main advantage of indoor production is that it gives superior control over temperature and lighting compared to outdoor production, which may result in higher productivity. The building used may be a greenhouse or just any industrial warehouse type of building of sufficient size. Using a greenhouse may reduce the need for artificial lighting. However, artificial lighting may still be needed to alleviate the reduction in growth during night-time and/or during the winter in the temperate climate zone. The use of greenhouses may, on the other hand, be associated with a more pronounced need for temperature control as greenhouses warm up during periods of intense sunlight and, in contrast, cool down during night and winter periods. 

If using warehouses, all the light necessarily has to come from artificial sources. Still, the need for temperature control may be less pronounced due to the insulating effect of the building walls. However, arrangements for heating the building during cold periods and cooling during warm periods should be made. However, it must be emphasized that even in indoor production systems relying entirely on artificial lighting, the microalgal cultures may heat up as a consequence of illumination. This is due to the fact that biomass absorbs some 94–97% of the incident long-wave (thermal) irradiance, and only 7% of the short-wave irradiance is utilized in photosynthesis add [90,91]. The remaining irradiance is absorbed as heat, and the effect on the temperature of the microalgal production system can be significant [41]. Therefore it may be necessary to install cooling devices in the photobioreactor, even in indoor production [89].

## 6. Light Availability and Mixing

Light availability can be one of the main constraints of large-scale cultivation of photosynthetic microalgae and cyanobacteria. With increasing algal density, light attenuates very quickly, the cells start to self-shade, and less light is available per cell. For example, Wang et al. [92] demonstrated that in a 20 cm deep pond, algal cells were fully illuminated in the first 3 days of cultivation, however as the algae density increased from day 4 to day 10, the light could only effectively penetrate 46% of the open-pond depth.

The aim is to make the surface-to-volume ratio very high while maintaining the cultivation system’s cost-effectiveness. In large cultivation ponds, solar radiation is typically the only source of light, and therefore, the light availability is dependent on climate, seasonality, and local weather [82]. When artificial light is supplemented with PBRs, the external light reaching algal cells inside the cultivation system depends on (1) PBR material, (2) biofouling inside, (3) light attenuation of medium or wastewater, (4) inoculum density, (5) PBR depth, and (6) mixing efficiency [23]. The depth of an open pond is generally less than 30 cm, and very high-density systems (e.g., thin-layer cascade PBRs) are less than 5 cm deep [93]. This is because ponds are generally less dense (lower algal density), and therefore they can be deeper as the light reaches further into the culture. 

Light and mixing are inherently tied. Optimized mixing of algal culture increases light availability per cell and nutrient and gas exchange. Mixing also prevents algae from settling, clumping, and creating unwanted films that can lead to bacteria accumulation and biofouling. Excessive mixing may generate harmful sheer stress or a turbulent environment that can damage sensitive cells. Extreme aeration can also produce unwanted foam and oxygen saturation, which may slow down photosynthesis. Suitable mixing is, therefore, critical for productive algae cultivation. On a large scale, mixing is generated by paddlewheels in ponds [94] and pumps in closed systems. As pumps are becoming more energy-efficient, their application increases even in ponds. Ponds with very large areas take advantage of flow rectifiers, baffles, delta wings, and deflectors [53] to optimize the mixing efficiency. Air and CO_2_ supplementation devices (carbonators), if placed strategically, can significantly improve mixing as well.

## 7. Monitoring

Monitoring a growing culture is necessary to evaluate progress, determine culture health, and establish harvesting plans. The variables that are commonly monitored are cell counts, optical density, dry weight, turbidity, pH, nutrient concentration, microscopic observations, and monitoring of target compounds such as lipids or pigments. The larger the cultivation system, the higher the chance of contamination and heterogeneity. Large systems may suffer from patchiness, localized fouling, and places where algae and cyanobacteria settle. The location of the sampling points is representative of the entire system, which is difficult to do on a large scale, so consistently sampling the same location (e.g., in the large pond) is key. Because monitoring can be expensive and time-consuming, scientists rely on established quick proxies such as optical density, pH, turbidity, and other sensors that can be automatically set up to sample. Automation can be expensive, and it still requires human oversight, but it significantly reduces the staff required, making the operation safer and potentially cheaper in the long run. Human input is still needed to calibrate and maintain the sensors, interpret the data, and troubleshoot the sensors. However, automation of the monitoring process can improve the capacity and sustainability of large-scale cultivation.

## 8. Production Modes

Algae cultivation methods differ based on their application and circumstances. The most prevalent approach is batch mode, where a significant portion of the culture is harvested and replaced with fresh medium, while some algae are left behind as an inoculum for the next batch. On the other hand, semi-continuous algae production involves more frequent harvests of only a portion of the culture. Continuous mode follows a similar principle, where harvesting is carried out continuously. Chemostats and turbidostats exemplify this mode of cultivation. In continuous cultivation, a specific parameter such as nitrate or turbidity is measured, and if the reading surpasses a predetermined threshold, only a small portion of the culture is replaced with fresh medium. Continuous cultures are typically more efficient in biomass production. For example, McGinn et al. [95] showed that algal productivities in chemostats were twice as high as the maximum rates observed in batch cultures. Continuous cultivation involves continuous harvesting and processing, which can be challenging on a large scale in terms of maintaining constant personnel oversight and covering equipment operational costs. Therefore, batch or semi-continuous modes are commonly used in large-scale cultivation systems. For example, Novoveska et al. [23] showed that frequent semi-continuous harvesting was more efficient than less frequent semi-continuous harvesting, likely due to improved mixing efficiency achieved through the harvesting process itself. Similar results were found on the small scale as well [96]. 

## 9. Contamination

### 9.1. Likelihood of Contamination

Biological pollutants become a significant constraint in mass cultivation, mainly in open systems like raceway ponds, but even enclosed PBR can suffer from contamination. Bacteria, zooplankton, (harmful) algae, and viruses are the main biopollutants that might constrain algae growth and impede the industrial process [97]. Contamination by biological pollutants is hard to avoid; open ponds evidently have a large open interface, while ‘closed’ photobioreactors have a weak spot because of the needed continuous aeration and light. Ecological principles such Theory of Island Biogeography [98] can be applied in this case as well because large area ponds enable the survival of many more contaminants when compared to small PBRs. Several mitigation methods have been proposed (e.g., filtration, chemical treatment with common pesticides such as Trichlorphon, Buprofezon, or other additives, and changing environmental conditions) [97]. Long-term open pond production can be sustained under extreme conditions such as high salinity, high alkalinity, low pH, etc., any condition that prevents competing organisms from thriving [99]. Biological contamination will always be a threat which implies that (safety) controls, monitoring, and early detection will always be needed for different applications and that adequate measures need to be taken for each particular use. 

### 9.2. Sterilization of Medium 

To avoid contamination, sterilization of growth media is required. For lab-scale work (≤20 L), autoclaving or filtration is usually the preferred method [13,100], while large volumes are generally treated using filtration and/or UV irradiation [89,101]. Common for filtration and UV irradiation is that there is no addition of chemicals and no toxic residues that negatively affect the microalgae [102]. However, filtration is not the same as sterilization, as small bacteria or viruses may pass through the filter, depending on the pore sizes. Autoclaving is an effective sterilization method, although it can raise the pH of seawater and cause precipitation of nutrients [103]. This may be countered by controlling pH during cultivation and adding sterilized nutrients after autoclaving. However, autoclaving is unrealistic in large-scale production, and UV radiation or ozonation is therefore widely used in these operations, including aquaculture, where pre-filtration is important for optimal effectiveness [101]. Filtration should primarily be used as a pretreatment before UV irradiation, as filtration alone has been shown not to be effective for longer-term cultivation [104].

### 9.3. Cultivation of Polyculture and Consortia

Large-scale commercial facilities generally focus on the mass cultivation of a single species with desirable traits. Maintenance of clean monocultures is, however, challenging and costly at large-scale. Cultivation of two or more species and the microbial communities at the same time may increase productivity via resource use efficiency and community stability [105,106]. Consortia occupy different functional niches and therefore maximize resource utilization. A community is also more stable under varying conditions. Lessons can be learned from agriculture, where herbicides, pesticides, and other exclusions are deployed to maintain monocultures, but these methods are expensive and may pose a risk to the environment. Wastewater treatment is one of the areas where consortia can be applied without sacrificing the product. 

In biofilms, the presence of bacteria and extracellular polymeric substances (EPS) is inevitable and wanted. Bacteria and EPS can create a positive environment on which to seed an algal biofilm [107]. There are two challenges with this interaction: (1) algal-bacterial presence could produce low-value products due to the contribution of bacterial biomass to the final harvested biomass, and (2) the process cannot be controlled easily and may take a long time to develop on its own. Modern techniques such as scanning electron microscopy (SEM) and catalyzed reporter deposition fluorescence in situ hybridization (CARD–FISH) can be used to observe the presence of microbes and EPS within the biofilm without major disturbances [107]. 

## 10. Harvesting and Dewatering 

### 10.1. Summary of Harvesting Methods

To harvest algal and cyanobacterial biomass from the medium, the culture is commonly concentrated through sedimentation, flocculation, flotation, filtration, and centrifugation. These multiple operations are time-consuming and often costly. In a recent techno-economic study of microalgae production for fuel, Davis et al. [37] estimated that biomass harvesting costs alone account for 21% of the total capital cost of an open pond cultivation system.

Harvesting biomass can be challenging due to the microscopic size of the cells. Some filamentous cells (e.g., *Arthrospira platensis*) can be easily harvested by filtration. Other very small cells (e.g., *Nannochloropsis* spp.) are more challenging to dewater. The selection of harvesting and dewatering methods depends on variables such as cell type, cell density, frequency of harvest, automation requirements, intended extraction and application process, and economic feasibility. While it is time-consuming, the initial passive (or flocculant-assisted) settling step can reduce some of the associated costs. 

There has been a boom in the development of harvesting and dewatering methods. There is a variety of techniques available to the consumer ranging from passive settling, the addition of flocculants and coagulants to create biomass aggregates and expedited settling, filtering (belt filters, membrane filters, cross-filters, oscillating filters, etc.), dissolved air and suspended air floatation, hydrocyclones, electrocoagulation, centrifuging and use of microfluidics to harvest, separate and sort cells [108,109]. Magnetically responsive nanoparticles were also successfully used to enhance microalgae growth, harvesting, and efficient separation of biologically active compounds on a small scale [110,111]. However, the cost-effective harvesting method remains one of the bottlenecks of large-scale cultivation. 

### 10.2. Milking Microalgae

As described above, harvesting microalgae is a major bottleneck in microalgal biomass production, especially due to the low biomass density in the culture medium. In many cases, the purpose of producing microalgae is to retrieve a secondary metabolite from the biomass. In those cases, it should be considered if it is possible to retrieve the secondary metabolite without the need to harvest the biomass destructively. This is referred to as milking microalgae. The analogy to dairy farming is obvious: In dairy farming, milk is produced by milking cows without the need to kill the cows. Similarly, in microalgae milking, secondary metabolites are retrieved without harvesting, killing, and extracting the microalgae, left in culture to continue production. The feasibility of milking microalgae has been demonstrated in the production of β-carotene from *Dunaliella salina* [112]. In this case, the microalgae in question is a species without a rigid cell wall [15], facilitating the extraction of β-carotene. However, microalgal milking has also been demonstrated for astaxanthin from *Haematococcus pluvialis* [113], a microalga with a rigid cell wall, especially in its red-cyst, astaxanthin-containing phase. For milking of microalgae, two-phase production systems need to be developed where the microalgae can easily be moved between the aqueous growth medium and the extraction solvent, that in most cases will need to be hydrophobic due to the hydrophobic nature of many secondary metabolites [112]. Furthermore, the extraction solvent has to be of low toxicity to the microalgae, and/or the contact time needs to be short to ensure the microalgae’s continued viability after milking [113].

### 10.3. Harvesting Biofilms 

The immobilized cultivation of microalgae biofilm has particular benefits in reducing the costs related to harvesting. In the biofilm system, algae are attached to a material surface and harvested by scraping. The harvested algae have a solid content of 10–20% (dry basis), which is similar to that of the post-centrifuged biomass [65].

Commercialization of biofilm technology for algal biomass production requires new strategies in order to obtain a high biomass production performance. Recently Zhang et al. [66] proposed the cultivation of algal biofilm using lignocellulosic materials. The results showed that lignocellulosic materials could be efficient carriers for low-cost cultivation of algal biofilm and the enhancement of biomass productivity. This is the first attempt to develop a new biofilm technology using lignocellulosic materials as biofilm carriers, such as pine sawdust, rice husk, sugar bagasse, and oak sawdust, all of which are environmentally friendly, cheap, and renewable with a wide range of distribution around the world. Moreover, after cultivation, algal biomass can be harvested together with the lignocellulosic materials. Amazingly, the harvested blend (mixture of the lignocellulosic materials and algal biomass) can be directly utilized as a feedstock for bioenergy conversion [66].

### 10.4. Utilization of Biomass without Harvesting

When producing microalgae for aquaculture live feed, there is usually no need for harvesting the microalgae. If the aquaculture organisms are filter feeders, which will usually be the case if microalgae are natural feed components for them, once a sufficient cell density has been achieved in the microalgal culture, the microalgae can be pumped directly into the aquaculture operation. A suitable cell density of microalgae will usually be more than 1 × 10^6^ cells mL^−1^. If the microalgal production takes place in batch culture (e.g., in plastic bags), the batches have to be emptied consecutively into the aquaculture production. If a continuous production, based on PBRs or similar, is used, a continuous feed regime can be established [89]. 

## 11. Environmental Risks of Large-Scale Cultivation

The environmental risks involving large-scale algae production are water quality and quantity, together with the impact on climate and biodiversity. The risks in closed and open cultivation systems are different. Concerning the quantity of water, large-scale systems consume a significant amount of water (fresh or salt water, depending on the species). Unlike closed systems, open systems are prone to significant evaporation (Table 1). Large-scale cultivation of microalgae usually requires the addition of nutrients such as phosphorus, nitrogen, silica, micronutrients, and trace metals [114], which may lead to an increase of nutrients in the effluent and receiving environments causing eutrophication. The aim is not to discharge valuable nutrients and either dose only sufficient nutrient concentration without any excess or recycle effluent and water [115]. An additional risk to water quality is the use of other chemicals such as pesticides, antibiotics, disinfectants, and other products used for cleaning purposes.

Large-scale cultivation may potentially impact the release of greenhouse gases (e.g., CO_2_, CH_4_, N_2_O). Although photosynthetic microalgae require and consume significant amounts of CO_2_ during growth, there may be a small release of greenhouse gases during the cultivation process, transport, and fertilizer application [116]. Depending on the type of system and the final product, the reduction in the emission of greenhouse gases varies. Open ponds, for instance, performed better than photobioreactors, along with the production of biodiesel and biomethane over ethanol production [117]. 

Large-scale cultivation may cause the release of the species that, in turn, may affect the receiving environment. The risk is more relevant nowadays due to the recent improvement in metabolic engineering, which leads to GM algae [118]. The effects of the possible release of genetically engineered organisms or elements need to be studied to determine the environment’s risks better [119]. The extent of the variant’s release and its fitness with regard to acquiring limited resources will determine the impact on the total biodiversity [120]. Bani et al. [121] clearly demonstrate that to maintain a specific inoculum, closed systems are preferred. Recent advances in molecular bioassessment could be used to assess microbial bioindices in receiving environments that indicate significant changes in the microbial community that could affect function [122]. In this sense, unintended biological and biochemical release from any bioprocess can be efficiently evaluated and environmental risks minimized. 

A life cycle assessment refers to the cradle-to-grave estimation of the environmental impacts of any given process in a standardized way [123]. It takes into consideration raw materials and resources, energy consumption, and emissions in a holistic approach from the raw materials to final products [124]. To be able to adequately use LCA approaches, a certain maturity of the technology is needed to limit significant variability due to assumptions/extrapolations due to limited data at the industrial scale [125]. The advantages of large-scale microalgae production of Spirulina compared to meat production are well-documented in terms of land and water needs and greenhouse gases emitted (less than 1%) [126].

## 12. Challenges of Large-Scale Cultivation

Scaling up the cultivation of microalgae and cyanobacteria poses numerous challenges, including the optimization of both upstream and downstream processes. The most common challenges are highlighted in Figure 2. The high cost of producing microalgae biomass remains the primary barrier to its commercial utilization. The estimated costs range from €290 to €570 per kg dry weight, depending on the specific production method employed [127]. Medeiros and Moreira [128] and Maroušek et al. [129] showed that the costliest aspects of microalgae are capital expenses for production units and operational cost production expenses related to liquid movement, lighting, temperature control, cleaning, and harvesting. Profitability for cultivation companies is typically achieved only at a very large scale. Additionally, the industry is burdened by investor mistrust following the rise and fall of the algal biofuel sector. The EU inception impact assessment on the blue bioeconomy [7] further identifies market gaps and an overall unfavorable business environment with some barriers to the growth of the algae sector in the EU. 

While lab-scale PBR can be operated by a single person, large-scale cultivation of microalgae requires a variety of expertise, e.g., engineers, technologist, chemists, and biologists, especially phycologists. The industry is confronted with a scarcity of algal expertise, with experienced phycologists in high demand and becoming increasingly rare. Yet, major universities now offer practical courses and programs to cultivate a new generation of algae experts and trained personnel, addressing the growing demand. The adoption of automation and cost reduction in labor will play a significant role in lowering cultivation costs. Despite this progress, social barriers persist, particularly in the acceptance of algae as a food source. The public remains hesitant to embrace new biomass feedstock like microalgae, although there is a growing trend towards vegetarian diets, with nearly 1.5 billion vegetarians worldwide [130]. Regulatory gaps remain a major obstacle as well.

The long-term success of large-scale cultivation systems depends on various factors. A crucial aspect is the availability of high-quality inoculum, or algae seed, to ensure reliable production. Not surprisingly, large-scale cultivation systems rely on smaller systems to generate enough inoculum and maintain backup inoculum. Operational challenges also include the adaptability of photobioreactors (PBR dimensions, mixing systems, baffles, etc.) and the need for customized medium, nutrients, and water recycling (with a focus on reducing culture media costs). Furthermore, there are several downstream processing considerations, such as selecting appropriate harvesting, cell disruption techniques, and extraction methods [131].

To overcome these challenges, various approaches have been proposed, including the utilization of continuous cultures, higher biomass densities, phycoprospecting, biorefinery concepts, genetic modifications (both indirect and direct), increased automation and computer control, and larger culture units. The latter aims to address the economy of scale and foster the professionalization of algal production. It has been estimated that with professionalization and scaling up of production, the cost of microalgal biomass in aquaculture operations could potentially decrease to as low as €43 per kg dry weight [127].

## 13. Conclusions

The cultivation of photosynthetic microalgae and cyanobacteria is gaining interest worldwide. These organisms provide societal benefits such as applications in food, feed, pharmaceutical, and cosmetic industries, as well as the wastewater treatment industry. In order to supply these industries with suitable quality and quantity of algal biomass, cultivation on a large scale is necessary. Commercial production of photosynthetic microalgae and cyanobacteria requires reliable cultivation on a large scale. The scale-up process is often expensive and frequently carries high risks. In this review, we aim to describe general design considerations, provide background information and discuss challenges associated with large-scale cultivation. We address all the aspects of large-scale cultivation, including motives for cultivation, species selection, types of cultivation (ponds, photobioreactors, and biofilms), water and nutrient sources, temperature, light and mixing, monitoring, contamination, harvesting strategies, and potential environmental risks. We also offer practical recommendations and discuss common challenges of large-scale systems associated with cost-effective design, operation, and maintenance, shortage of experienced staff (especially phycologists), and social barriers, such as wider acceptance of algae as a food source and regulatory gaps.

## Figures and Tables

**Figure 1 marinedrugs-21-00445-f001:**
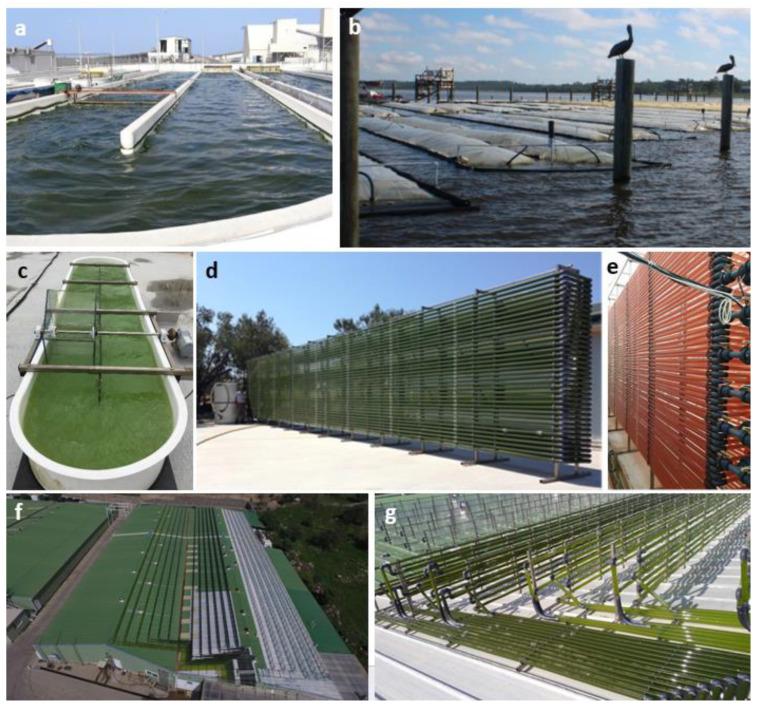
Examples of large-scale cultivation of photosynthetic microalgae and cyanobacteria. (**a**) Open pond (Israel) (photo credit Soren Nielsen) (**b**) Floating offshore closed PBRs with capacity 8000 L each at Algae Systems LTD (USA) (photo credit Lucie Novoveská) (**c**) 1000 L pond (Turkey) (**d**–**g**) Large-scale 40,000 L tubular PBRs at Akvatek (Turkey).

**Figure 2 marinedrugs-21-00445-f002:**
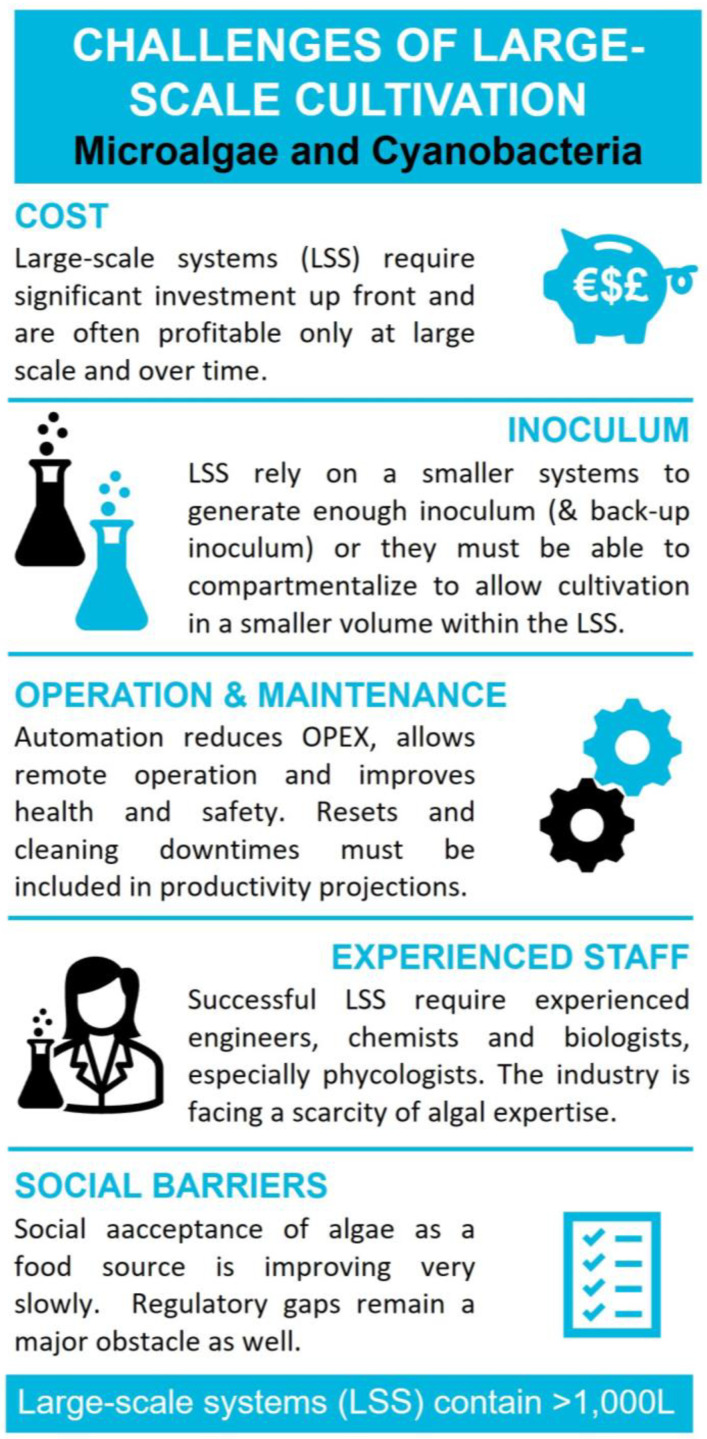
Informatics highlighting the most common challenges associated with large-scale cultivation systems.

**Table 1 marinedrugs-21-00445-t001:** Comparison of large-scale cultivation methods.

Critical Factors	Open Ponds	Closed PBRs	Biofilms
Land area required	High	Medium	Medium
Predator/debris exclusion	No	Yes	No
Thermal regulation	No	Medium	No
CO_2_ feed efficiency	Low	High	Low
Evaporation	High	Low	Medium
Mixing energy required	Medium	High	Medium
Additional light source	None	Yes	None
Volume throughput	High	Medium	Low
Cell density (for harvest)	Low	Medium	High
Contamination risk	High	Low	High
Capital cost	Low	High	Low

## Data Availability

Not applicable.

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
