# Peer review of "Overview and Challenges of Large-Scale Cultivation of Photosynthetic Microalgae and Cyanobacteria"

_marinedrugs, 2023, doi:10.3390/md21080445_

Round 1
Reviewer 1 Report
This paper systematically introduced the necessity of large-scale cultivation of microalgae, the selection of algal species, culture methods, water and nutrient sources, temperature regulations, light availability and mixing etc. The manuscript was well written and organized and could be accepted after minor revision.
1. What is the meaning of “Age of Algae”, please verify it.
2. Table 1 should be changed to a three-line form.
3. The summary of Figure 2 is not very accurate, please summarize it again or delete the figure and replace it with words.
Author Response
We would like to thank the reviewer for their time and feedback which certainly improved our manuscript.
- What is the meaning of “Age of Algae”, please verify it.
“The Age of Algae and Novel compounds” represents an era in which interest in algae dramatically increased, algae came to the spotlight, there are more companies cultivating algae, there are more articles about algae and overall more public interest in algae and products containing algae.
- Table 1 should be changed to a three-line form.
Table 1 was changed to a three-line form. Contamination risk was added to the Table upon request of Reviewer 3.
- The summary of Figure 2 is not very accurate, please summarize it again or delete the figure and replace it with words.
The article is very word-heavy and it would benefit from some visual aids. We have reworded the text and made significant changes to the infographics.
Reviewer 2 Report
Revision of the review article titled:
Overview and challenges of large-scale cultivation of photosynthetic microalgae and cyanobacteria
For the MDPI Journal: Marine Drugs
General observations:
The review article is generally well written and quite interesting although it could be more focused and informative.
I kindly suggest some major revisions that, in my opinion, would improve the scientific value of the article.
The title suggests that review deals with both microalgae and cyanobacteria, but there is only small comparison of the two (lines 39-44). Please add paragraph which would explain similarities and differences in the large scale cultivations of these two groups of photosynthetic microorganisms. For example, A. platensis requires pH = 9.0 while C. vulgaris requires pH = 7.0. How this affects each media composition, nutrients removal, and how to maintain pH in large systems etc. Please consider adding more discussion on this topic (species dependent PBR differences, downstreaming differences, cell disruption differences etc.).
I suggest that you insert a new Table with the overview of the most important species used in the current long term cultivations (in the industry and/or public sector), for biomass, fine chemicals, bio fertilizers or bio oils production. In the table you can also add columns with the preferred PBR, the biggest constrains and pluses for each strain cultivation. This, in my opinion, would add great value in the informative aspects of this work.
Please add Conclusion section. Although it’s not obligatory for this type of article, it would add depth and importance to the work if you list the most important findings that resulted from this literature overview.
Line by line comments:
Lines 90-99: Please explsin importance, if any, of these ingredients to fish nutrition, (for example Iodine) and what can be done to enrich the biomass during cultivation….
Line 125: Please check Italics in algae species names.
Line 135: You mention the Novel food rule (law) for the marketing of the whole cell algae from 15th May of 1997. Please explain it to the readers and add number and citation of the EU Commission Regulation, if applicable.
Line 172-173: Please reformulate RUBISCO sentence to have more sense.
Line 212-213: Macroalgae cultivation is unrelated to the topic. Please consider removing the sentence.
Line 252: Word “although” appears three times in this line. Please reformulate.
Line 265: Please add more references and values of PBR volumetric productivities. What about cyanobacteria? Is it really always in 10 - 12 g/m3/day range, as you stated?
Line 329-330: Is there any other disadvantage? Maybe dead or anoxic zones in horizontally placed large closed plastic bags? Please reconsider.
Line 365: What are the biggest advantages of natural sea water? For example, do we need to add micronutrients or Ca and Mg salts to seawater? Please reconsider.
Line 397: Are there any constrains in later application of the algal biomass produced on municipal wastewater? Can it be used readily as human food? Please explain.
Line 429: Please mention the importance of natural mechanism of cooling in open ponds (latent enthalpy of water vaporization). Or state otherwise if this is not important/correct.
Line 476: Please explain in more detail. For example, you mention 5 cm depth of thin layer cascade and 30 cm depth of open pond. What part of incoming light penetrates these depths?
Line 488: Is there any negative effect of intense mixing to algal cell? Sheer stress or similar, please explain.
Line 507: Please replace “person power” with manpower.
Line 542: “chemical additives” - please name some of the most common chemicals used for grazer prevention with references.
Line 603: Is there any application of this technique in the large scale cultivations, that you know of? If not please consider removing.
Please consider some of the recent articles dealing with long term cultivation of different algal species that are not mentioned in the paper and cite them (when you find it suitable):
DOI:10.3390/app10051725;
DOI:10.3390/pr9081333;
DOI:10.1016/j.algal.2021.102500;
DOI: 10.3390/pr9081326;
DOI:10.1007/s10811-019-01873-y;
References: Please replace references number 38, 58 and 78 with more recent ones (published after January 2000).
Reviewer 3 Report
The paper is well written and needs revision.
In the "Abstract" section, the authors did not indicate the scientific novelty of review article.
Line 157, could you give GM method example for adjusting the chloroplast antenna size.
For Table 1, you should add “contamination” as a critical factor
In section 3.2, please give info related to the efficiency of heat and mass transfer for PBRs.
In section 3.3, in which sector are the algal biofilm systems used more?
In section 4.2, could you give examples for agricultural fertilizers, please?
In section 4.3, for the best algal growth what should be the ratio of C:N:P in WWT?
In section 11, mention life cycle analysis (LCA) as a solution proposal for the management of environmental risks.
Author Response
Reviewer 3
We would like to thank the reviewer for their time and feedback which certainly improved our manuscript.
- In the "Abstract" section, the authors did not indicate the scientific novelty of review article.
The scientific novelty lies in a fact that majority of the co-authors have practical experience with large scale cultivation and are industry experts which is extremely rare and valuable. We have added a sentence to the abstract: “Importantly, we also present practical recommendations and ….”
- Line 157, could you give GM method example for adjusting the chloroplast antenna size.
The text has been modified and sentence added: Adjusting the chloroplast antenna size is one of the most common GM methods to sustain high biomass productivity at large scale especially for products requiring higher light intensities [30,31].
For Table 1, you should add “contamination” as a critical factor
We agree with the suggestion, the risk of contamination was added to Table 1.
- In section 3.2, please give info related to the efficiency of heat and mass transfer for PBRs.
Thank you for the comment. To provide a detailed summary for heat and mass transfer is beyond the scope of our manuscript. We have addressed this point in Section 3 and added a sentence: "PBRs are built with high ratio of surface area to culture volume to maximize illumination, however, the same principle allows a heat loss in the absence of light. Understanding time-dependent heat balance within the PBR is critical for accurate productivity predictions [41]."
- In section 3.3, in which sector are the algal biofilm systems used more?
We have reworded the sentences: “Algal biofilm reactors were first reported in the 1980s used for removing nitrogen and phosphorus from municipal wastewater. To date, progress has been made in using microalgae biofilm for wastewater treatment and biomass production.”
- In section 4.2, could you give examples for agricultural fertilizers, please?
Agricultural fertilizer is any brand of fertilizer used by farmers in any country. The authors felt that giving examples would be country specific and perhaps it can be viewed as promoting in nature.
We have added a sentence: “It is generally found that the use of agricultural fertilizers, such used in farming in general, yields production rates of biomass equal to or even better than traditional media [72–74], while at the same time providing equal or better production yields of the required end product [75].”
- In section 4.3, for the best algal growth what should be the ratio of C:N:P in WWT?
The C:N:P ratio varies highly with the type of wastewater, for example, wastewater from a brewery is very high in C (COD/BOD) but very low in N & P, while agricultural runoff is richer in N & P. Different microalgal species also prefer different C:N:P ratios (Table 2 in https://doi.org/10.1016/j.scitotenv.2020.142168). For these reasons, there is no single best ratio. Also, it is a common practice to enrich or dilute wastewater to make it suitable for specific organism.
- In section 11, mention life cycle analysis (LCA) as a solution proposal for the management of environmental risks.
A paragraph was added at the end of section 11:
"A life cycle assessment refers to cradle-to-grave estimation of the environmental impacts of any given process in a standardized way (ISO 14040:2006). It takes into consideration raw materials and resources, energy consumption, emissions in a holistic approach from the raw materials to final products. To be able to adequately use LCA approaches, a certain maturity of the technology is needed to limit significant variability due to assumptions/ extrapolations due to limited data at the industrial scale (Nilsson et al. 2019 ). The advantages of large-scale microalgae production of Spirulina compared to meat production are well-documented in terms of land and water needs and greenhouse gases emitted (less than 1%) (Tzachor 2022)."
Round 2
Author Response
We would like to thank the reviewer again for their time and effort in improving our manuscript.
We have addressed the 2 minor points as requested:
1) Additional information was added to Section 6 Light and Mixing (current line 578) addressing light penetration, algal density and typical depths of ponds/PBRs.
2) Morillas-Espana et all reference was added to Section 4.3. Use of Wastewater as Nutrient Source and Section 6 Light and Mixing - impact of climate on algal growth (thank you for highlighting this manuscript to us).
Thank you